# Assessment on Physical and Rheological Properties of Aged SBS Modified Bitumen Containing Rejuvenating Systems of Isocyanate and Epoxy Substances

**DOI:** 10.3390/ma12040618

**Published:** 2019-02-19

**Authors:** Zhelun Li, Xiong Xu, Jianying Yu, Shaopeng Wu

**Affiliations:** 1State Key Laboratory of Silicate Materials for Architectures, Wuhan University of Technology, Wuhan 430070, China; lizhelun@whut.edu.cn (Z.L.); wusp@whut.edu.cn (S.W.); 2School of Civil Engineering and Architecture, Wuhan Institute of Technology, Wuhan 430074, China

**Keywords:** SBS-modified bitumen, rejuvenating systems, physical properties, viscous-elastic temperature, rutting factor, vibration noise consumption

## Abstract

Styrene–butadiene copolymer (SBS)-modified bitumen (SMB) is widely applied in pavement construction. With yearly services, many SMB wastes urgently need to be reclaimed for repaving roads based on the objectives of environmental protection, landfill saving, as well as resource utilization. The present work is focused on the investigation of the physical and rheological properties of aged SMB incorporated with rejuvenating systems consisting of fluid catalytic cracking slurry (FCC slurry), C12–14 aliphatic glycidyl ether (AGE), diphenylmethane diisocyanate (MDI), and other additives. The rejuvenating systems containing the main components of 10% FCC slurry, 10%FCC/3%AGE, and 10%FCC/3%AGE/1% MDI were respectively recorded as R_a_, R_b_, and R_c_. The results indicate that both R_b_ and R_c_ have obvious workability that make contributions for improving comprehensive physical properties while slightly reducing the softening point, which were also proven to be effective for the re-rejuvenation of re-aged binder. The higher viscous-elastic temperature caused by the agglomeration of binder molecules in aged SMB could be dropped to a lower value with rejuvenating systems, while improving the low-temperature crack resistance. With the use of the R_b_ and R_c_ rejuvenating systems, the high-temperature deformation resistance of aged SMB fell, approaching the performance of fresh SMB. Vibration noise consumption could be improved for aged SMB incorporated with R_b_ and R_c_ in the form of viscous loss, while the effects for re-aged SMB containing the same rejuvenating systems were weakened but still effective.

## 1. Introduction

Along with the booming development of highway construction in China, tri-block styrene–butadiene copolymer (SBS)-modified bitumen is widely applied in high-class pavement due to its advantageous high- and low-temperature performance, driving comfort, smooth surface, and abrasive resistance [1,2,3,4]. Despite that, during the in-service period, a series of physical and chemical changes resulting in the performance deterioration of SBS-modified bitumen (SMB) occur under the comprehensive effect of the natural environment such as heat, UV, oxygen, and rainwater [5,6,7,8]. Furthermore, a great deal of waste SMB is constantly produced and causes huge resource and landfill waste as well as environmental hazards [9,10]. Accordingly, the problem of high-quality rejuvenation for waste SMB urgently needs to be solved.

The aging of SMB includes not only oxidation and poly-condensation of base bitumen, but also the oxidative degradation of SBS, which differs from the aging of base bitumen [11]. Owing to the oxidative degradation of SBS, the road performances of SMB after long-term aging significantly drop to hardly meet standardized requirements [12,13,14,15]. Thus, the consideration regarding the comprehensive performance recovery of SMB should be taken for both aged bitumen and aged SBS. As of now, some publications have reported that the rejuvenation of aged SMB can be realized by adding fresh SMB and/or rich aromatic oils [16,17,18,19,20]. For instance, Gong et al. investigated the physical and chemical properties of aged SMB mixed with bio-oil derived from biodiesel residue, and discovered that the bio-oil could be used to achieve the goal of rejuvenating aged SMB and enhance its physical properties, while mitigating the highly-oxidized components aggregated and dispersing the asphalt molecules [21]. Chen et al. conducted a study using waste edible vegetable oil to rejuvenate aged asphalt binder containing SBS by testing the physical and rheological properties, and found that waste edible vegetable oil is suitable for the recycling of aged SBS modified asphalt with the result of better performance recovery [22].

Although these kinds of physical components have some advantages in restoring most of the properties of aged SMB, the performance recovery is only for aged bitumen binder and not aged polymer. In fact, some publications have reported that the oxidation and degradation of SBS would simultaneously occur during the oxidative aging step, with the destructed polymer pieces forming with oxygen-containing groups such as hydroxyl and carbonyl groups [23,24,25,26]. Considering the use of those groups, we propose an in-situ chemical reaction to amend the molecular structure of SBS polymer in order to improve the properties of bitumen binder. In view of this emerging idea, the novelty differs from other reports in trying to use the reactive components to make a partial connection among degraded polymer, and the high aromatic mixtures to adjust the chemical composition of aged virgin bitumen are both involved.

This present work aims to solve the issue of high-quality recycling of aged SMB by using the selected rejuvenating system consisting of fluid catalytic cracking slurry (FCC slurry), C12–14 aliphatic glycidyl ether (AGE), diphenylmethane diisocyanate (MDI), and other additives. The physical and rheological properties of aged SMB incorporating the rejuvenating systems were systematically investigated, and the re-rejuvenation effects of the selected rejuvenating system are discussed and analyzed.

## 2. Experimental

### 2.1. Raw Materials

The used SBS modified bitumen (SBS/bitumen, 5/100) was obtained in the laboratory by mixing linear SBS (1301, S/B = 30/70) and base bitumen (SK-70). FCC slurry was selected to improve the chemical components of aged bitumen binder. The relative mass percentages in saturates, aromatics, resins, and asphaltenes were 8.82%, 75.23%, 7.93%, and 8.02%, respectively. AGE is a low-viscosity end-epoxy molecule that can penetrate into the binder and disperse the agglomeration formed from aging. MDI was selected to limit the dropping high-temperature properties of rejuvenated SMB through the chemical consolidation reaction.

### 2.2. Aging Procedure of SMB

The evenly dispersed SMB specimens were standardly prepared for pressure aging vessel (PAV) aging. The aging experiments were as follows in order of the thin film oven test (TFOT, referring to ASTM D1754) and the PAV test (referring to ASTM D6521) [27], the lab conditions of which were 163 °C × 5 h and 100 °C × 2.1 MPa × 20 h (in air atmosphere) to simulate the short-term mixing and paving process and the long-term working aging of SMB, respectively.

### 2.3. Rejuvenation for Aged SMB

The aged SMB specimens above were quantitatively weighed and shifted to an agitated vessel for hot mix rejuvenation. Until the stable temperature of 150 °C was reached, the low-speed agitation started, while 10% FCC slurry, 3% AGE and other low amounts of additives with or without 1% MDI were respectively mixed into the binder for 20 min to prepare three kinds of rejuvenated SMB. The rejuvenating systems containing the main components of 10% FCC slurry, 10%FCC/3%AGE, and 10%FCC/3%AGE/1% MDI are respectively abbreviated as R_a_, R_b_, and R_c_ for reference in this manuscript.

### 2.4. Tests for Physical Properties

The physical properties of various SMB specimens as mentioned above, included viscosity at 135 °C, softening point, penetration at 25 °C, and ductility at 5 °C were measured according to ASTM D4402, ASTM D36, ASTM D5, and ASTM D113, respectively [28,29,30,31]. Additionally, the experimental results of the primary physical properties of fresh and PAV-aged SMB are presented in Table 1.

### 2.5. Tests for Rheological Properties

The rheological characteristics of the various SMB specimens mentioned above were tested using a dynamic shear rheometer (DSR, MCR101, Anton Paar, Graz, Austria). The tests for rheological properties were conducted under strain-controlled conditions at 10 rad/s with the heat rate of 2 °C/min between the temperature ranges of −10 and 80 °C. During the test, plates with an 8 mm diameter and a 2 mm gap below 30 °C and plates with a 25 mm diameter and a 1 mm gap above 30 °C were respectively used. According to the manufacturing specifications, related rheological parameters such as elastic modulus (G′), viscous modulus (G″), complex modulus (G*), phase angle (δ), and rutting factor (G*/sin δ) were simultaneously obtained to assess the rheological properties of rejuvenated SMB.

## 3. Results and Discussion

### 3.1. Effect of Rejuvenating Systems on the Physical Properties of Aged SMB

Table 2 presents the related experimental results regarding the physical properties and recovery rates of aged SMB incorporating rejuvenating systems. From the table, with rejuvenating systems, the improvement on the ductility and penetration of aged SMB was clearly visible, while the softening point and viscosity both dropped to some extent. The differences in the physical properties of rejuvenated SMB containing R_a_, R_b_, and R_c_ were easily observed in that R_b_ and R_c_ had significant advantages in making contributions to the ductility recovery, penetration recovery, and viscosity reduction of aged SMB—except for the softening point—in comparison with R_a_. The explanation for the results is that the epoxy components play a vital effect in diffusing and dispersing the agglomerated substances of the aged binder. Meanwhile, compared with R_b_, the slight weakness of R_c_ for improving the physical properties is the reaction consolidation caused between MDI and aged SMB. 

Furthermore, the data show that the use of R_b_ and R_c_ for aged SMB promoted ductility recovery rates approaching 76.6% and 67.1%, penetration recovery rates of 112.5% and 95.9%, the viscosity indexes were 37.5% and 49.2%, and the softening point retention rates were 71.6% and 79.1% while the results of rejuvenated SMB containing R_a_ were respectively 37.0%, 70.8%, 44.9%, and 79.7%. The data shown here indicate that the rejuvenating systems containing AGE with or without MDI can work better to both restore the physical properties of aged SMB and obtain rejuvenated SMB with various behaviors suitable for re-application.

### 3.2. Effect of Rejuvenating Systems on the Physical Properties of Second Aged SMB

Figure 1, Figure 2, Figure 3 and Figure 4 display the physical properties of ductility, softening point, penetration, and viscosity of aged SMB containing R_b_ and R_c_, respectively. It is clear that rejuvenated SMBs containing R_b_ or R_c_, after second aging, almost lost their low-temperature ductility, accompanied by a rise in the softening point. When R_b_ and R_c_ were selected again for the rejuvenation of second-aged SMB, the ductility increased by 23.7 and 20.2 cm with the recovery rates of 67.7% and 57.6%, while the softening point decreased by 5.1 °C and 3.4 °C with retention rates of 74.5% and 80.6%. The results indicate that the rejuvenating system can, to some degree, restore the low-temperature properties of second-aged SMB and depress the significant deterioration of high-temperature properties.

Meanwhile, the penetration increased by 25 dmm and 21 dmm with recovery rates of 97.9% and 83.3%, which illustrates that the use of these kinds of rejuvenating systems was still effective in restoring the penetration of second-aged SMB. Lastly, the results of the viscosity index (46.1% and 65.2%) show that the viscous behavior of second-aged SMB were improved using R_b_, and increased with the addition of MDI. In other words, the use of R_c_ was effective in restoring the low-temperature properties and limited the sharp fall of high-temperature properties for the second-aged SMB. To conclude, both R_b_ and R_c_ had good working abilities for recycling re-aged SMB in order to achieve better properties.

### 3.3. Effect of Rejuvenating Systems on the Viscous-Elastic Behavior of Aged SMB

The viscous-elastic characteristics of aged SMB with incorporated rejuvenating systems are presented in Figure 5. As depicted in Figure 5a, the values of G′ and G″ of fresh SMB after aging increased, and the viscous-elastic temperature shifted to a higher level from 14.6 °C to 24.2 °C, indicating that the deformation resistance of fresh SMB increased during the aging period, causing the viscous portion (G″ after 14.6 °C) of fresh SMB to tend to the harder portion before 24.2 °C. From Figure 5b, it can be clearly seen that the values of G′ and G″ of rejuvenated SMB were to some extent lower than the aged one, and the viscous-elastic temperature of aged SMB decreased with the addition of rejuvenators. Compared with R_a_, the exhibited viscous behaviors of R_b_- and R_c_-rejuvenated SMB were more obvious, the viscous-elastic temperatures of which decreased to 8.1 °C and 12.1 °C, respectively. The findings illustrate that rejuvenating systems such as R_b_ and R_c_ provide the benefit of activating the hard components in aged SMB to the viscous components, and improve the low-temperature crack resistance. The reason for the higher viscous-elastic temperature of rejuvenated SMB containing R_c_ depends upon the chemical reactions between the isocyanate groups and the formed reactive groups (e.g., –OH and –COOH) in aged binder [9].

### 3.4. Effect of Rejuvenating Systems on the Physical Properties of Aged SMB

The loss tangent (tan δ) is a vital indicator used to depict the damping loss of materials. Figure 6 displays its tendency and changing law for various bitumen specimens containing fresh, aged, and rejuvenated types. Useful and interesting information obtained from the figure is that the loss tangent of fresh SMB decreased with aging, however, to a certain degree, the loss tangent of aged SMB increased with rejuvenation. In other words, with the hardening of fresh SMB during the aging period, the damping loss was be reduced on the basis of elastic energy storage, and when the aged SMBs were rejuvenated with R_a_, R_b_, and R_c_, the damping loss was recover to a higher level. These results indicate that the vibration consumption of aged SMB containing rejuvenating systems may give them potential for use in reducing tire–ground noise. Differences in the damping loss of rejuvenated SMB incorporating R_b_ and R_c_ can mainly be attributed to the fact that the chemical reactions between isocyanate and aged SMB cause the partial hardening of the mixes, leading to a decrement of the viscous loss.

### 3.5. Effect of Rejuvenating Systems on the Complex Modulus and Phase Angle of Aged SMB

The complex modulus and phase angle of aged SMB with rejuvenating systems are displayed in Figure 7. In the temperature region between −5 and 30 °C, the complex modulus and phase angle of fresh SMB respectively increased and decreased after aging. With rejuvenation, the complex modulus recovered to that of the original or lower level of fresh SMB, while the phase angle increased to near to or greater than the level of the fresh SMB. Compared with R_c_, R_b_ had a more obvious role in promoting the temperature sensitivity of aged SMB. All of the results demonstrate that the rejuvenating systems, particularly R_b_, can activate the transfer of the hard components to the soft components in aged SMB, allowing superior rheological characteristics to be obtained.

### 3.6. Effect of Rejuvenating Systems on Rutting Factor of Aged SMB

Figure 8 shows the rutting factors of aged SMB incorporating R_a_, R_b_, and R_c_ with increasing temperature. The temperature points inserted in this figure refer to the requirements of the SHRP specification. It can be seen that the rutting factor of fresh SMB visibly increased after aging, while the rutting factor reduced with rejuvenation, among which rejuvenated SMB containing R_b_ exhibited the greatest decrease. The results obtained indicate that the rejuvenation systems, especially R_b_, had significant effects in decreasing the high-temperature characteristics of aged SMB. The findings show that depending upon the low-viscosity epoxy components in R_b_, it can easily permeate into the aged binder and disperse the agglomerated substances formed by aging. Additionally, from the inserted figure, it can be seen that the rutting factor of rejuvenated SMB containing R_c_ was somewhat higher in comparison with that containing R_b_, demonstrating that the chemical reaction between R_c_ and aged SMB can cause an increase in the rutting factor, and thus improve the high-temperature deformation resistance of rejuvenated binder.

### 3.7. Effect of Rejuvenation System on the Phase Angle of Second-Aged SMB

Due to better rejuvenation of aged SMB with R_c_ according to the discussions above, the research on the rheological properties of second-rejuvenated SMB is continued in this section. Figure 9 shows the effect of R_c_ on the phase angle of second-aged SMB. By comparing the phase angle trends of second-aged and rejuvenated SMB, it can be clearly obtained that the obvious differences of the phase angle were observed before approximately 50 °C (namely, the consistently higher phase angle of second-rejuvenated SMB), while the value of second-rejuvenated SMB was more similar to the fresh one. The results indicate that the flow behaviors of second-aged SMB can be restored by this kind of rejuvenation system. Regarding the first-rejuvenated SMB containing R_c_, four phase angle-temperature points (−5, 5, 25, and 60 °C) were selected to comparatively analyze the rejuvenation ability of R_c_. At these four positions, the second-rejuvenated SMB consistently exhibited a lower rheological characteristic when compared with the first-rejuvenated one, which indicates that the rejuvenation ability of R_c_ was weakened for second-aged SMB. To summarize, this type of rejuvenation system is still effective and useful to recover the rheological properties, and manifests better rejuvenation ability for second-aged SMB.

### 3.8. Effect of Rejuvenation System on the Damping Loss of Second-Aged SMB

The two temperatures of 5 °C and 60 °C were selected as references to investigate the low- and high-temperature viscous damping loss of first- and second-rejuvenated SMB containing R_c_, the data of which are presented in Table 3. As can be seen, regardless of whether the temperature was at 5 or 60 °C, the values of second-rejuvenated SMB were relatively lower when compared with the first-rejuvenated one. The results indicate that the viscous damping loss and deformation degree of second-rejuvenated SMB cannot recover to those of the first rejuvenation, but that second-rejuvenated SMB can also show relatively good rheological properties. The explanation for this is that the re-aging and re-rejuvenation bring about the agglomeration of higher-weight components and reaction-consolidation substances formed in the asphalt binder, and thus affect the consumption of vibration energy through the viscous loss of second-rejuvenated SMB.

## 4. Conclusions

This research aimed to achieve the high-quality performance recovery of aged SMB using rejuvenating systems consisting of FCC slurry, AGE, MDI, and other additives. The physical and rheological properties of rejuvenated SMB were systematically investigated and evaluated. Some worthy and interesting results are summarized below:Both R_b_ and R_c_ rejuvenation systems of had significant advantages in contributing to improving the physical properties, including the ductility, penetration, and viscosity of aged SMB, while slightly reducing the softening point. Meanwhile, these kinds of rejuvenating systems were proved to still be applicable in the re-rejuvenation of re-aged binder.The results of viscous-elastic temperatures indicate that oxidative aging promotes the hardening caused from the agglomeration of binder molecules for fresh SMB, which leads to a sharp increase of the viscous-elastic temperature. However, with the rejuvenating systems, particularly R_b_, the viscous-elastic temperature of rejuvenated SMB could be somewhat reduced to improve the low-temperature crack resistance.The obtained results on rutting factors indicate that the rejuvenating systems, especially R_b_ and R_c_, are harmful to the high-temperature deformation resistance of rejuvenated SMB, but their performance level can still be close to that of fresh SMB with changing temperature.The results of damping loss demonstrated that vibration consumption for noise was improved for aged SMB with incorporated R_b_ and R_c_ in the form of viscous loss, and furthermore, the effects for re-aged SMB using the same rejuvenating systems were weakened but still effective.We recommend the use of rejuvenating systems containing epoxy—with or without isocyanate—for recycling aged SBS asphalt binder for application in the mid-temperature region. The disadvantage of rejuvenated binder lies in its poorer high-temperature workability. Despite that, more considerations regarding the performance restoration of degraded polymer in binder should be examined in future work.

## Figures and Tables

**Figure 1 materials-12-00618-f001:**
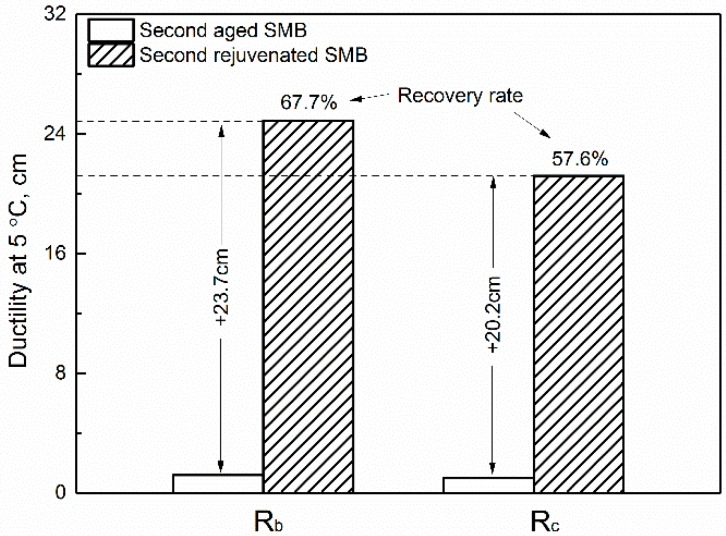
Effect of rejuvenating systems on the ductility of second-aged SMB.

**Figure 2 materials-12-00618-f002:**
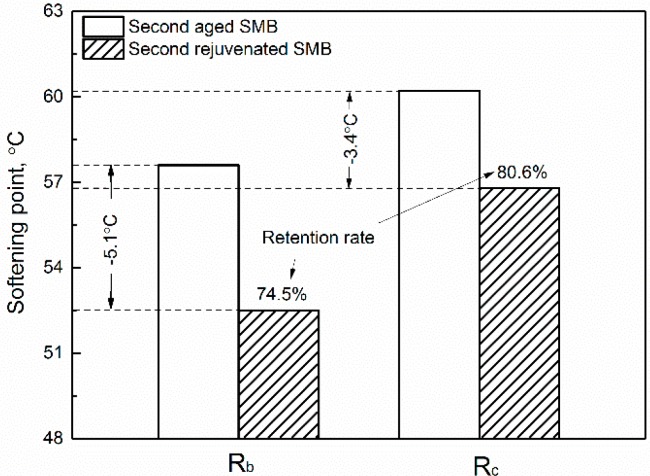
Effect of rejuvenating systems on the softening point of second-aged SMB.

**Figure 3 materials-12-00618-f003:**
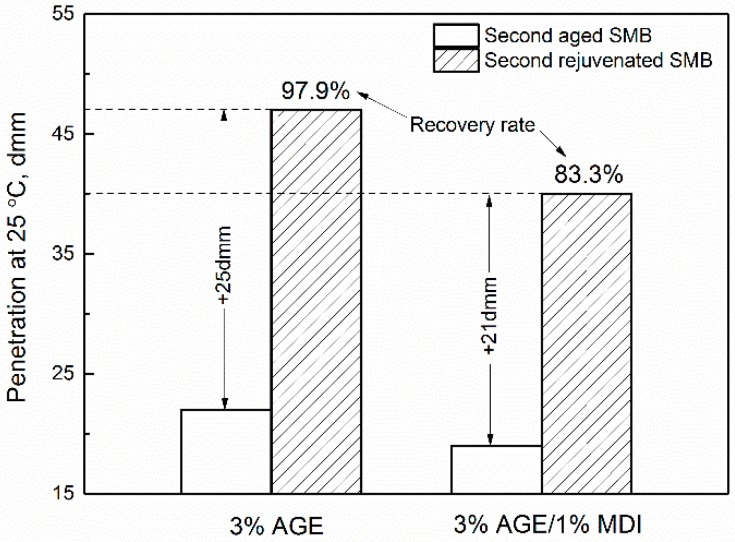
Effect of rejuvenating systems on the penetration of second-aged SMB.

**Figure 4 materials-12-00618-f004:**
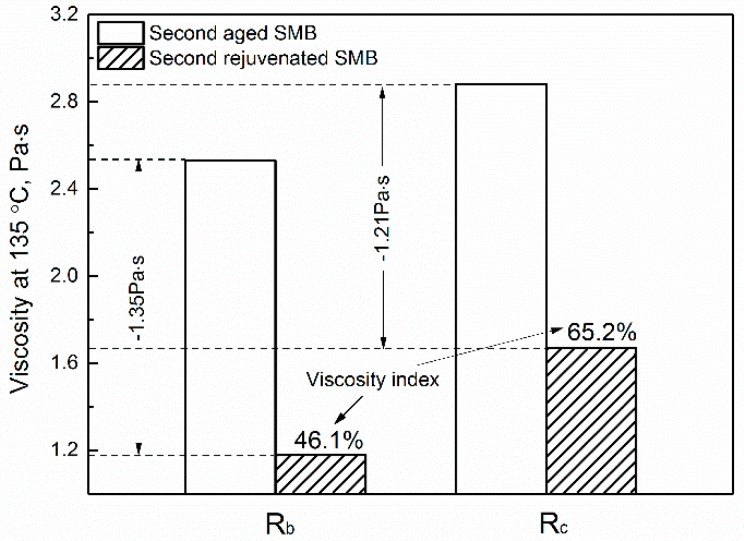
Effect of rejuvenating systems on the viscosity of second-aged SMB.

**Figure 5 materials-12-00618-f005:**
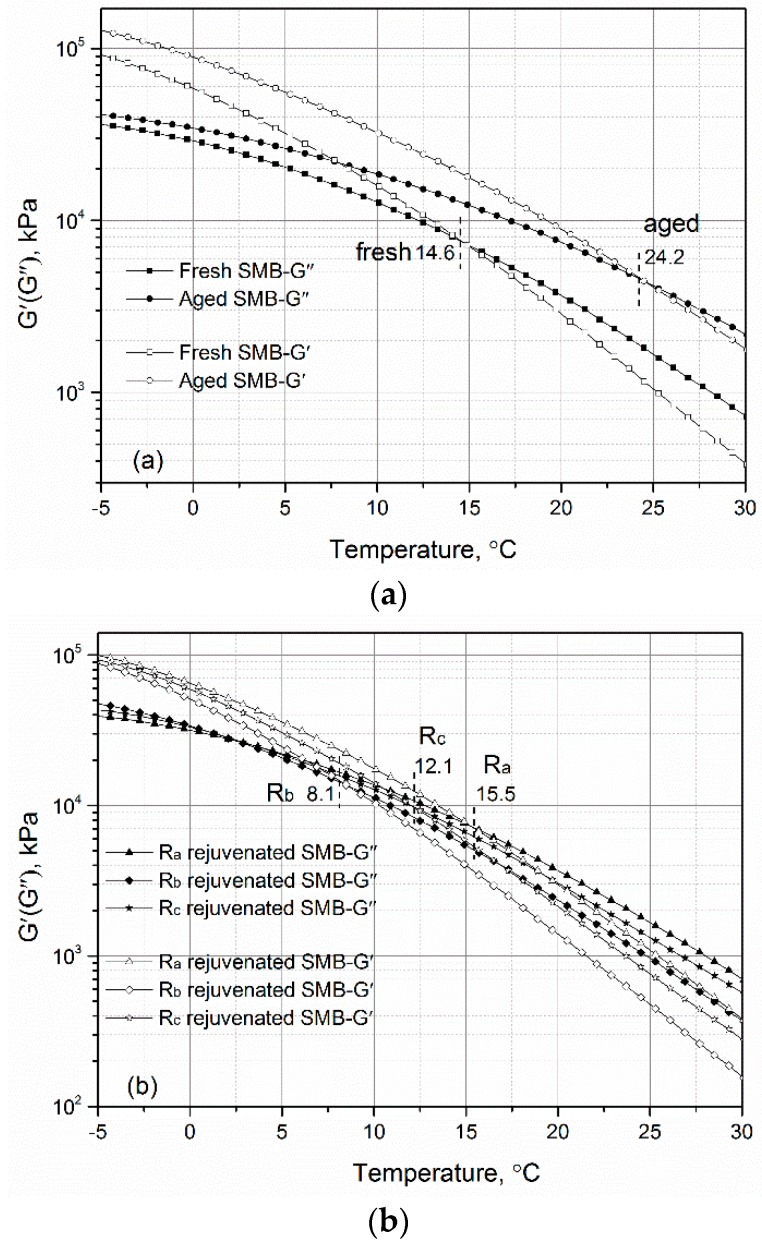
Viscous-elastic characteristics of (**a**) fresh and aged SMB and (**b**) rejuvenated SMB.

**Figure 6 materials-12-00618-f006:**
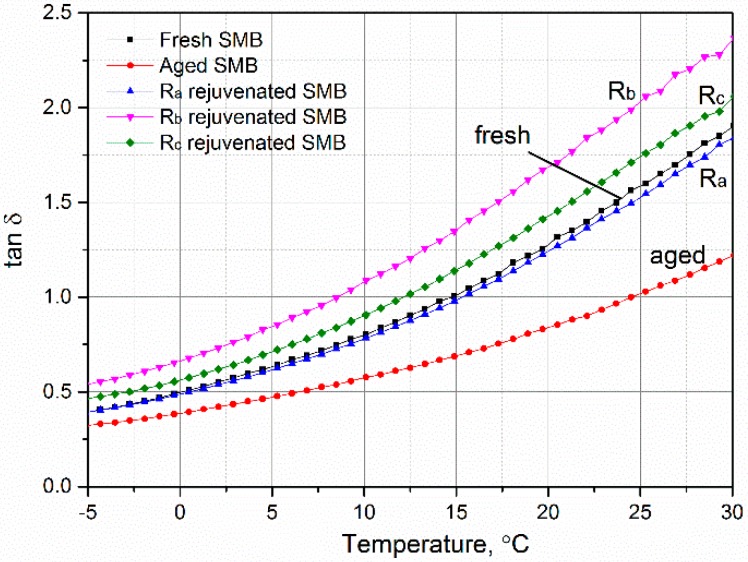
Effect of rejuvenating systems on the loss tangent of aged SMB.

**Figure 7 materials-12-00618-f007:**
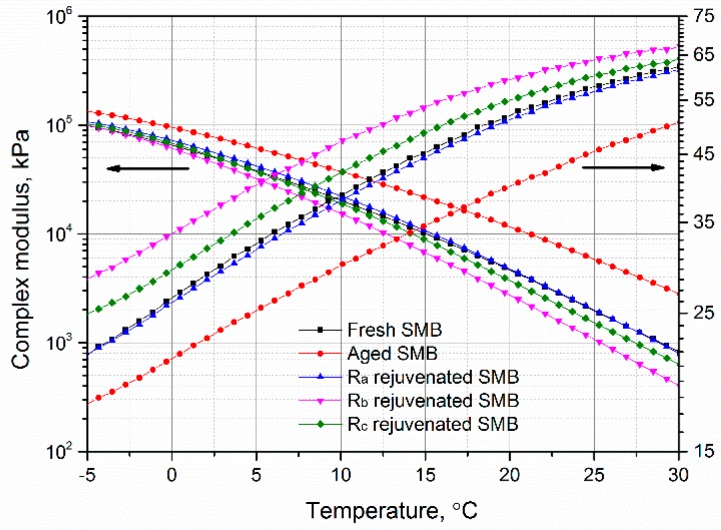
Effect of rejuvenating systems on the complex modulus and phase angle of aged SMB.

**Figure 8 materials-12-00618-f008:**
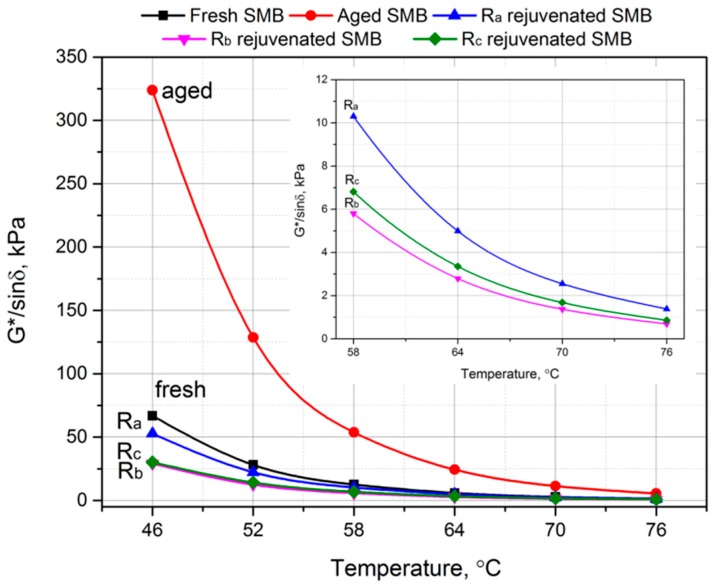
Rutting factor (G*/sin δ) of fresh, aged, and rejuvenated SMB versus temperature.

**Figure 9 materials-12-00618-f009:**
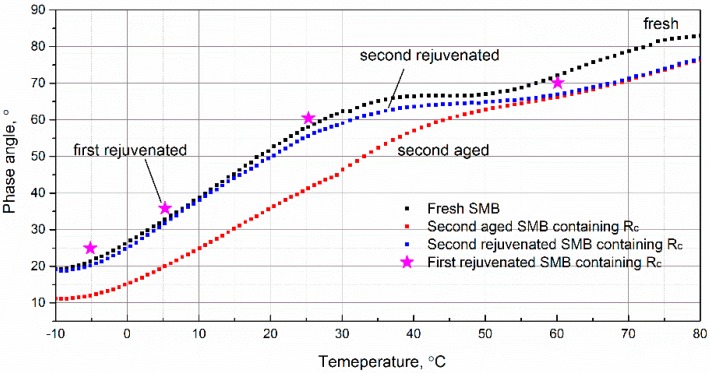
Effect of rejuvenating system on the phase angle of second-aged SMB.

**Table 1 materials-12-00618-t001:** Primary physical properties of styrene–butadiene copolymer modified bitumen (SMB) before and after pressure aging vessel (PAV) aging.

Technical Index	Fresh SMB	PAV-Aged SMB
Ductility at 5 °C (cm)	36.8	1.2
Softening point (°C)	70.5	63.1
Viscosity at 135 °C (Pa·s)	2.56	2.85
Penetration at 25 °C (dmm)	48	17

**Table 2 materials-12-00618-t002:** Physical properties and recovery rates of aged SMB incorporating rejuvenating systems.

Technical Index	Aged SMB	Rejuvenated SMB			Equation
		R_a_	R_b_	R_c_	
Ductility at 5 °C (cm)	1.2	13.6	28.2	24.7	-
Softening point (°C)	63.1	56.2	50.5	55.8	-
Penetration at 25 °C (dmm)	17	34	54	47	-
Viscosity at 135 °C (Pa·s)	2.85	1.15	0.96	1.26	-
Ductility recovery rate (%)	-	37.0	76.6	67.1	$\mathrm{PCR}=\frac{P_{R}}{P_{F}}\times 100\%$
Softening point retention rate (%)	-	79.7	71.6	79.1	
Penetration recovery rate (%)	-	70.8	112.5	95.9	
Viscosity index (%)	-	44.9	37.5	49.2	

**Note:** In the equation, PCR refers to the performance recovery ratio for rejuvenated binder; P_R_ refers to the value for physical properties of rejuvenated binder; and P_F_ refers to the value for the physical properties of fresh binder.

**Table 3 materials-12-00618-t003:** Loss tangent of first- and second-rejuvenated SMB containing 10%FCC/3%AGE/1% MDI (R_c_) at 5 °C and 60 °C.

Samples	δ, °		Tan δ	
	5 °C	60 °C	5 °C	60 °C
First-rejuvenated SMB containing R_c_	35.8	70.1	0.72	2.76
Second-rejuvenated SMB containing R_c_	31.8	66.9	0.62	2.34

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
