# Peer review of "Assessment on Physical and Rheological Properties of Aged SBS Modified Bitumen Containing Rejuvenating Systems of Isocyanate and Epoxy Substances"

_materials, 2019, doi:10.3390/ma12040618_

Round 1

Reviewer 1 Report

Please consider conducting further characterization on your "aged binder" if you want to really want to study the effects of your additives in rejuvenating the aged SMBs. For example,  measuring the asphaltenes size (molecular weight), before and after aging and studying the effect of your rejuvenators would be beneficial to the reader. 

I am not convinced that what you claim are rejuvenators are actually rejuvenating the asphalt or just simply softening it. 

Author Response

Thanks for your kind suggestions. What you advised is way useful to improve our manuscript. The characterizations like the asphaltene size (molecular weight), before and after aging, and the effect of rejuvenators, are also most important to study the effect of rejuvenating additives on SMBs. In the current manuscript, we aim to centrally investigate the physical and rheological properties of rejuvenated SMBs, and the chemical characterizations related to FTIR, GPC, and other measurements are shown in our previous studies. Some citations are inserted in this manuscript.

Again, many appreciated thanks for your careful reading and checking. Your comments are pretty good for our future work.

Reviewer 2 Report

The authors present a study looking at the effects of 3 rejuvenating  formulations on the physical properties of a parent styrene-butadiene modified bitumen (SMB).  The rationale for the study is sound, with the need to reduce environmental and financial impacts by developing more robust materials for pavement purposes.

The methodology is straightforward with standard methods, and the conclusions that were reached are sound.  It appears that the Rb and Rc formulations might be promising for use in further product development

A few suggestions:  

1) line 11 change "has" to "is"

2) Sections 2.5 and 3.1 are redundant so I suggest removing section 3.1

3) the modulus symbols on lines 107, 153, 156, 157 are missing characters (i.e. (G') and (G") ).

4) line 224 spacing between the numbers and oC symbols

Author Response

1) line 11 change "has" to "is"

R: Be done. Thank you for your careful reading.

2) Sections 2.5 and 3.1 are redundant so I suggest removing section 3.1

R: Sorry for my carelessness. The text of section 3.1 is missing whilst adjusting the format according to the requirement of MATERIALS. In the revision, the replacement of the correct content is conducted.

3) the modulus symbols on lines 107, 153, 156, 157 are missing characters (i.e. (G') and (G") ).

R: Sorry for my carelessness while changing the format of this manuscript.

4) line 224 spacing between the numbers and oC symbols

R: Sorry for my carelessness. The spacing in need is inserted. Thanks for your helps on the quality of this paper.

Reviewer 3 Report

The paper is generally well written and the production process is thoroughly described. Results are well presented. But there are a number of comments that should be corrected to make the paper publishable.

1) In “Introduction” section novelty of the paper should be clarified. What is your research offering the field that hasn’t been offered before?

2) P3L94 “penetration at 25°C” but Y-axes in fig 3 indicated as “penetration at 135°C”. What was the test temperature?

3) Section 3.1 should be presented in the Experimental” section.  

4) P3 L107 and L118, and further through the text: “elastic modulus (G ), viscous modulus (G  )” provide correct designations.

5) Figs 1-4 show results only for the Rb and Rc, why the results for the Ra are not shown in the figs?

6) Generally, section 3 should be “Results and discussion” section instead of only “Results” section. Systematic analysis of existing literature regarding the properties of the same materials can improve impact of the paper.

7) For use the produced materials in highway constructions it is necessary to study performance properties (mechanical and tribological properties, stability and other) of the composites. In the paper there is no information about these properties. 

Author Response

1) In “Introduction” section novelty of the paper should be clarified. What is your research offering the field that hasn’t been offered before?

R: Thanks for your suggestion. Actually our novelty of the paper aims to achieve the objective of the partial reconstruction of degraded SBS in asphalt binder. The emerging mind for rejuvenating aged SBS modified bitumen is including two aspects of degraded SBS and aged virgin bitumen. To clear the novelty of the paper, the content is supplemented in Introduction, stating that “In view of the emerging mind, the novelty differs from other reports is summarized that trying to use the reactive components to make the partial connection among degraded polymer, and the high aromatic mixtures to adjust the chemical composition of aged virgin bitumen are both involved.”

2) P3L94 “penetration at 25°C” but Y-axes in fig 3 indicated as “penetration at 135°C”. What was the test temperature?

R: Thank you for your careful reading. The test temperature for penetration is 25°C. Sorry for the error in my plotting. You know that the bitumen at 135°C approaches to the viscous flow, which cannot be tested for penetration.

3) Section 3.1 should be presented in the Experimental” section. 

R: Thanks. Section 3.1 has been deleted in the revised version.

4) P3 L107 and L118, and further through the text: “elastic modulus (G ), viscous modulus (G  )” provide correct designations.

R: The revisions related to what you suggested are conducted for the entire text.

5) Figs 1-4 show results only for the Rb and Rc, why the results for the Ra are not shown in the figs?

R: Thank you for your checking. I am sorry for the losing section 3.1 while adjusting the format according to MATERIALS’s requirement. In the original version, the results for Ra are conducted for discussion in comparison with that of Rb and Rc. The relevant contents have been supplemented in section 3.1 in our revised version. I am pretty appreciating your careful reading for my paper again.

6) Generally, section 3 should be “Results and discussion” section instead of only “Results” section. Systematic analysis of existing literature regarding the properties of the same materials can improve impact of the paper.

R: Thanks for your comments. The section tittle has been revised to “Results and discussion”. And then, some citations related to the literatures are displayed in Introduction to state the properties of the rejuvenated bitumen materials. The losing section supplemented in section 3.1 analyzes the differences of physical properties for three kinds of rejuvenated SMB, comparatively. Thank you again. The systematic analysis of existing literature regarding the properties of the same materials will be enhanced in our future papers.

7) For use the produced materials in highway constructions it is necessary to study performance properties (mechanical and tribological properties, stability and other) of the composites. In the paper there is no information about these properties.

R: Good question and promising advice. The performance properties you suggested are pretty important in highway construction, which will be further studied in our future work. Additionally, the rejuvenated binder will be blended with aggregates to prepare mixtures, and the engineering performances will be centrally discussed.

Round 2

Reviewer 1 Report

Please work on the grammar on the abstract.

Author Response

Thanks for your advice. The abstract has been proofread and revised further.

Reviewer 3 Report

Thank you for answer. Tha paper can be published in the journal. 

Author Response

Thanks.